# Evaluation of Interference Analysis from 5G NR Networks to Aeronautical and Maritime Mobile Systems in the Frequency Band 4800–4990 MHz

**Alexander Pastukh** [1,]*  **and Vladislav Sorokin** [2]

1   Radio Research and Development Institute, 105064 Moscow, Russia
2   The General Radio Frequency Center, 117997 Moscow, Russia
*   Correspondence: apastukh@lenta.ru

**Abstract:** The current deployment of 5G NR in the frequency band 4800–4990 MHz in multiple countries near the coastlines indicated that there might be a potential risk of harmful interference impacting aeronautical and maritime systems located in international airspace and international waters. This raised numerous concerns about whether 5G NR system rollouts in that band need to be restricted by the power flux density limits created at the border of the territorial waters, which equals 12 nautical miles from the coastline. This work contains a case study based on the example of the Pacific region, where a simulation of aggregate interference from 5G NR base stations and user equipment deployed in the cities near the coastline using Monte Carlo analysis was conducted. The results of the study show that no harmful interference to the aeronautical and maritime services operating in international airspace and waters is expected even when the aircraft or vessels are located at the border of the territorial airspace and waters from the coastline. Therefore, no restriction to the 5G NR deployment in the coastline regions in the frequency band 4800–4990 MHz is required.

**Keywords:** 5G NR; AMS; MMS; Monte Carlo analysis; frequency sharing; spectrum engineering; electromagnetic compatibility; aeronautical services; maritime services; interference analysis

## 1. Introduction

Today, 5G NR technologies play one of the most important roles in the sustainable development of countries, and different types of enhanced applications such as IoT, smartphone connectivity, industrial applications, smart cities, vehicle communications, and many others require high network capacity. To fulfill the requirements of all emerging 5G applications, an extra spectrum is needed for capacity expansion. According to numerous reports, including GSMA, to meet the demands of future 5G applications in a long-term period, an extra 2 GHz of the spectrum might be required to be allocated for 5G [1]. At present, the following bands are allocated for international mobile telecommunications (IMT), and some of these bands are already in use for 5G and some of them are still being studied for compatibility issues. Current bands identified for IMT are presented in Table 1, and the bands are divided into three categories: low bands, middle bands, and high bands. The low bands are good for providing higher coverage since the propagation conditions are much better in these bands, particularly because they allow us to improve residential connections, provide smart energy services, and provide rural coverage. Middle bands are good for providing urban coverage for different types of services, including smartphone connectivity, vehicle communications, transport infrastructure, etc. High bands allow for providing services in small areas where extremely high capacities are required, such as large-scale events (e.g., sports games, concerts, exhibitions, and so on).

**Table 1.** List of the bands currently identified for IMT.

| Low Bands | Mid Bands | High Bands |
| --- | --- | --- |
| 450–470 MHz 470–608 MHz 614–698 MHz 694–960 MHz | 1427–1518 MHz 1710–2025 MHz 2110–2200 MHz 2300–2400 MHz 2500–2690 MHz 3400–3600 MHz 3600–3700 MHz 4800–4990 MHz | 24,250–27,500 MHz 37,000–43,500 MHz 45,500–47,000 MHz 47,200–48,200 MHz 66,000–71,000 MHz |

The frequency bands 5G NR were split into frequency range one and frequency range two (FR1 and FR2). Frequency range one works at sub-6-GHz or below the 6 GHz band, whereas frequency range two covers the millimeter waves [2–4].

This study focuses particularly on the 4800–4990 MHz band for the reason that this band has potential problems with international compatibility with incumbent services which require interference analysis. During the World Radiocommunication Conference in 2015 (WRC-15), organized by the International Telecommunication Union (ITU), a significant number of countries added their name to footnote 5.441B of the Radio Regulations, which identified the frequency band 4800–4990 MHz for International Mobile Telecommunications (IMT) [5–7]. Additionally, 3GPP has designated the 4400–50,000 MHz band for 5G NR standard under the prefix n79 frequency, which includes the 4800–4990 MHz band. The strongest interest to use this band for 5G NR applications was shown by China, Russia, Japan, Brazil, Vietnam, South Africa, Nigeria, and many other countries. Overall, 40 countries are included in current footnote 5.441B of the Radio Regulations, thus the 4800–4990 MHz developed an interest from a significant number of countries.

This, however, has raised a question of protection of the aeronautical and maritime mobile services operating in this band, in particular those aeronautical and maritime receivers that are located in international airspace or waters. The frequency range of 400–4990 MHz is allocated on a primary basis in all three ITU regions to the mobile service. Under the mobile service allocation, systems and networks operating in the aeronautical and maritime mobile service comprise stations for broadband, airborne, or shipborne data links to support remote sensing and stations of aeronautical mobile telemetry.

These systems may experience interference from the aggregate interference of 5G NR deployed in the cities near the coastline, especially since many of the countries that showed interest in the 4800–4990 MHz frequency band have access to the oceans, where a lot of aircraft and vessels may use that band while being in international airspace or waters at a different distance from the coastlines. In particular, the Unites Nations Convention on the Law of the Sea (UNCLOS) and the International Civil Aviation Organization (ICAO) define those international waters and airspace as beginning at a distance of 12 nautical miles from the coast; additionally, UNCLOS defines an exclusive economic zone (EEZ) where each sovereign state has exclusive rights regarding marine activities, which may stretch up to 200 nautical miles [8,9]. Thus, interference levels from 5G NR networks deployed in the coastline areas need to be calculated to understand whether 5G NR rollout would cause harmful interference.

This type of scenario is pretty challenging due to the fact that aggregate interference from the technologies such as cellular networks have a long-term impact because the networks are gradually getting deployed, slowly increasing external noise near the deployed areas. Apart from that, studies to assess the impact of an entire 5G NR network should take into account the varying nature of a 5G NR network, in particular the power control of user equipment, network loading factors, TDD factors, beamforming of base stations, etc. It is important to note that it is nearly impossible to measure aggregate interference levels in field tests before the deployment of the networks since field tests would involve only a few numbers of interfering devices which will not have any significant impact on

victim receivers. Therefore, computer simulations are the only possible way to estimate the aggregate interference level from the cellular systems deployed in large areas.

The objectives of this research were achieved by doing simulations of aggregate interference from the 5G NR networks deployed in the cities near the coastline. For example, in this study, the Far Eastern region of Russia was chosen. This example is good for several reasons; first, it has some big cities located near the coastline, and second, the clutter height of buildings in this area is not high compared to countries such as China, for example, and thus the aeronautical and maritime stations are effectively less shielded from the interference of 5G NR networks deployed inside urban areas. Therefore, one of the variants of the worst-case scenario is considered.

## 2. State of the Art

At present, there are no studies published on that matter. Several other studies were conducted in ITU-R, but they have not been published in any relevant journal or conference paper [10]. This is mainly related to the fact that in the past years, when cellular technologies were coming to different frequency bands, there were never any compatibility issues between cellular technologies and aeronautical/maritime mobile services. Therefore, this work provides a novel approach in evaluating sharing conditions between 5G and aeronautical and maritime mobile services, which includes defining the proper scenario conditions for the simulations as well as the methodology of Monte Carlo analysis, which takes into account 5G NR base station beam position changes and the movements of aircraft and vessels. This allows us to perform close-to-real interference analysis which would allow us to understand whether 5G might be deployed in the 4800–4990 frequency band.

## 3. Materials and Methods

### 3.1. Study Assumptions and Scenario

To estimate interference from 5G NR to the aeronautical and maritime systems, a Monte Carlo analysis simulation with the SEAMCAT tool was used [11,12]. Since the wanted link of aeronautical and maritime systems fluctuates depending on antenna types and the distance between wanted link transmitters and receivers, in simulations the protection criterion of the aeronautical and maritime mobile systems was interference-to-noise ratio (I/N). An increase in receiver effective noise of 1 dB would result in significant degradation in communication range. Such an increase in effective receiver noise level corresponds to an (I + N)/N ratio of 1.26 or an I/N ratio of about −6 dB. When multiple potential interference sources are present, protection of the AMS and MMS systems requires that this criterion is not exceeded due to the aggregate interference from multiple sources [13].

Each simulation consisted of 10,000 events. During the simulations, the aircraft and vessels were placed at a distance of 12 nautical miles (22 km), 80 nautical miles (150 km), and 200 nautical miles (370 km). The 12-nautical-mile distance was considered as the worst-case scenario, whereas 80 and 200 are more likely distances in practice due to the EEZ areas. The 5G NR networks were located in 5 cities in the coastline area—Vladivostok, Artyom, Nakhodka, Ussuriysk, and Arsenyev. Figure 1 shows the scenario of the aeronautical mobile station (AMS) and maritime mobile station (MMS) located at a distance of 12 nautical miles from the coastline.

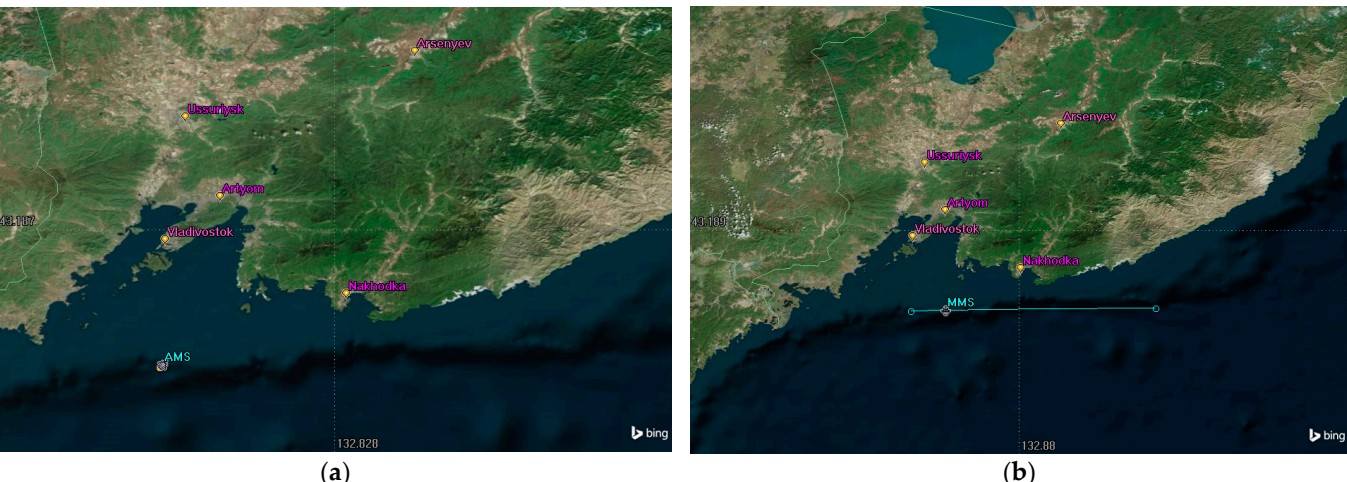

(**a**)                                              (**b**)

**Figure 1.** Scenario for the victim receiver station is located at 12 nautical miles from the coastline:
(**a**) aeronautical mobile station at 12 nautical miles from the coastline, and (**b**) maritime mobile station
moves along the border of territorial waters.

In each city, a 5G NR network was simulated and according to Recommendation
ITU-R M.2101 consisted of 19 tri-sector BS with 3 UEs at each sector, and thus interference
was calculated from 285 sectors overall. It should be noted that many cities have much
more IMT BS; however, they do not operate at the same frequency as AMS receivers
simultaneously since the spectrum in this band is divided between the mobile network
operators and inside each operator's deployments to avoid intercell interference, and base
stations do not reuse the same spectrum as that in the neighboring cell. Additionally, in
many of these areas, operators rely on the other bands as well, and therefore 57 sectors per
city is sufficient to estimate the interference level [14]. The typical deployment topology of
the 57 sectors networks provided in Figure 2. The red dots represent base stations and the
yellow dots represent users.

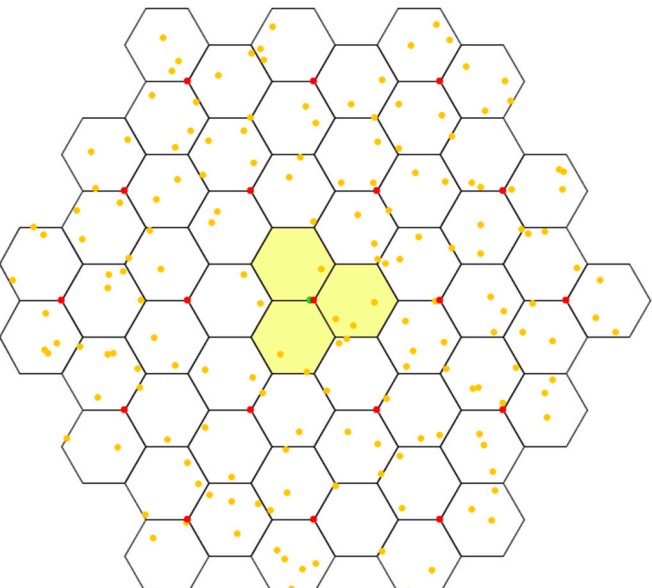

**Figure 2.** Typical 5G NR urban network topology.

*3.2. Characteristics of 5G NR Networks*

Tables 2 and 3 provide deployment-related and user equipment characteristics that
were used in the simulation, taking into account that for the 4800–4990 MHz range, contigu-
ous coverage is not expected in this frequency range in rural areas, and the study analyzes

interference only from urban deployment. These characteristics were derived from 3GPP specifications as well as from the Working Party 5D that is responsible for studying 5G technology within the ITU.

**Table 2.** Deployment characteristics of 5G NR networks in the 4800–4990 MHz frequency band.

| Parameter | Value |
| --- | --- |
| Cell radius | Typical cell radius 0.4 km urban |
| Base station antenna height | 20 m urban |
| Sectorization | 3 sectors |
| Frequency reuse | 1 |
| Typical channel bandwidth | 40 or 80 or 100 MHz |
| Network loading factor (base station load probability X%) | 20%, |
| TDD/FDD | TDD |
| BS TDD activity factor | 75% |

**Table 3.** Deployment characteristics of 5G NR networks in the 4800–4990 MHz frequency band.

| Parameter | Value |
| --- | --- |
| Indoor user terminal usage | 70% |
| Indoor user terminal penetration loss | Rec. ITU-R P.2109 |
| User equipment density for terminals that are transmitting simultaneously | 3 UEs per sector |
| UE height | 1.5 m |
| Average user terminal output power | Use transmit power control |
| Typical antenna gain for user terminals | −4 dBi |
| Body loss | 4 dB |
| UE TDD activity factor | 25% |
| Power control model | Refer to Recommendation ITU-R M.2101 Annex 1, Section 4.1 |
| Maximum user terminal output power, PCMAX | 23 dBm |
| Power (dBm) target value per RB, P0_PUSCH | −92.2 |

Nowadays, the development of antenna technologies for 5G NR sub-6-GHz has led to more effective channels as well as a reduction in interference. A lot of publications have been devoted to that issue [1–3]. In this study, active antenna systems (AAS) with beamforming were simulated, and non-AAS systems were not considered since they are not expected to be implemented in 4800–4990 MHz. The AAS antenna array model is determined by the array element pattern, array factor, and signals applied to the array system. One difference between a passive antenna system (e.g., based on Recommendation ITU-R F.1336) and an active advanced antenna system (AAS) is that for the AAS, the unwanted (out-of-the-block) emission sees a different antenna behavior compared to the wanted (in-block) emission. A 5G NR system using an AAS actively controls all individual signals being fed to individual antenna elements in the antenna array to shape and direct the antenna emission diagram to a wanted shape, e.g., a narrow beam towards a user. Table 4 shows the beamforming characteristics of the 5G NR base station in the frequency 4800–4990 MHz frequency band.

Based on the characteristics above, Figure 3 shows composite sub-array IMT antenna pattern in the 4800–4990 MHz frequency band that was used in the study.

When simulating beamforming, three spatially directive signals were transmitted pointing to each UE within the sector, and the UEs were distributed randomly within the sector, equally distributing the power of each beam between the users [15–17]. The beamforming antenna is based on an antenna array and consists of many identical radiating elements located in the *yz*-plane with a fixed separation distance (e.g., l/2), all elements having identical radiation patterns and "pointing" (having maximum directivity) along the *x*-axis. Figure 4 presents 3D modeling of the beamforming antenna pattern that was used in the simulations.

**Table 4.** Deployment characteristics of 5G NR networks in the 4800–4990 MHz frequency band.

| Parameter | Value |
|---|---|
| Antenna pattern | Recommendation ITU-R P.2101 |
| Element gain (dBi) | 6.4 |
| Horizontal/vertical 3 dB beam width of single element | 90° for H<br>65° for V |
| Horizontal/vertical front-to-back ratio (dB) | 30 for both H/V |
| Antenna polarization | Linear ± 45° |
| Antenna array configuration (Row × Column) | 4 × 8 elements |
| Horizontal/Vertical radiating element/sub-array spacing, $d_h$ /$d_v$ | 0.5 of wavelength for H, 2.1 of wavelength for V |
| Number of element rows in sub-array, $M_{sub}$ | 3 |
| Vertical radiating element spacing in sub-array, $d_{v,sub}$ | 0.7 of wavelength of V |
| Pre-set sub-array down-tilt, $\theta_{subtilt}$ (degrees) | 3 |
| Array Ohmic loss (dB) | 2 |
| Conducted power (before Ohmic loss) per antenna element/sub-array (dBm) | 28 |
| Base station horizontal coverage range (degrees) | ±60 |
| Base station vertical coverage range (degrees) | 90–100 |
| Mechanical down-tilt (degrees) | 10 |

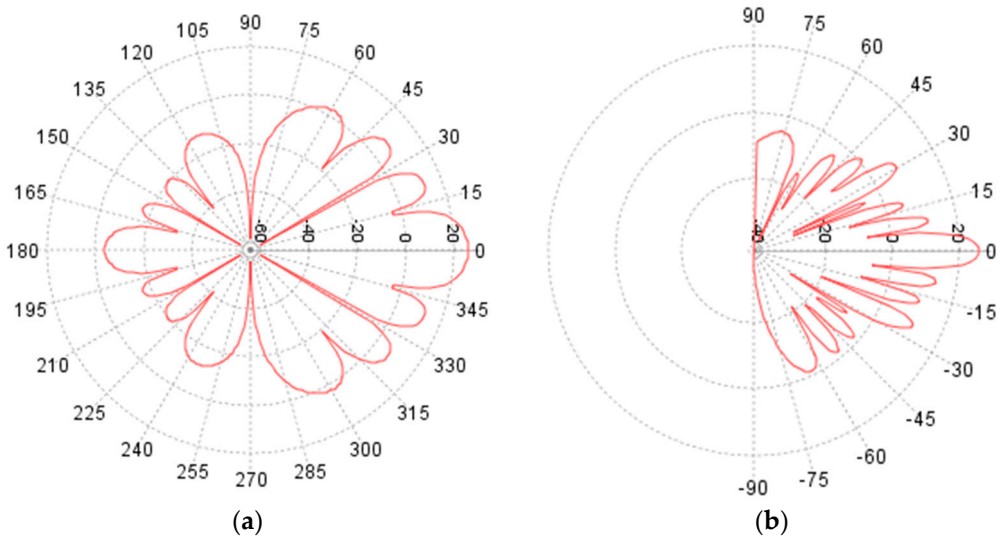

(**a**)          (**b**)

**Figure 3.** Antenna pattern of 5G NR base station: (**a**) azimuth plane and (**b**) elevation plane.

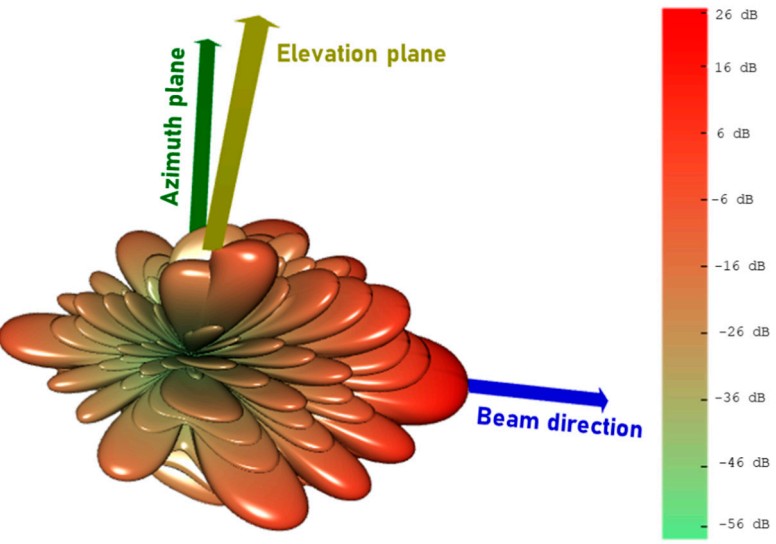

**Figure 4.** 3D modeling of the 5G NR base station beam.

### 3.3. Characteristics of Aeuronautical Mobile Service

Two types of AMS receivers were considered in the studies: the one with an omnidirectional antenna (System 1) and the one with a directional antenna (System 2). These parameters are described in Recommendation ITU-R M.2116. Table 5 presents characteristics of the AMS that were used in the simulations.

**Table 5.** Characteristics of the aeronautical mobile systems used in the simulations.

| Parameter | System 1 Airborne | System 2 Airborne 1 | System 2 Airborne 2 |
| --- | --- | --- | --- |
| Tuning range | 4400–4990 | 4400–4990 | 4400−4990 |
| Power output | 45 | 30–43 | 30–43 |
| Bandwidth (3 dB) | 1 | 5/0.008 | 5/0.008 |
| Noise figure | 3.5 | 6 | 6 |
| Thermal noise level | −110.5 | −103/−131 | −103/−131 |
| Antenna type | Omnidirectional | Directional | Directional |
| Antenna gain | 3 | 14 | 14 |
| 1st sidelobe | N/A | −1 | −1 |
| Polarization | Vertical | Vertical | Vertical |
| Antenna pattern | N/A | Uniform distribution Rec. ITU-R M.1851 | Uniform distribution Rec. ITU-R M.1851 |

The directional antenna of AMS was pointed to the nadir; however, it should be noted that in practice, the direction of it will be to the opposite direction of the coastline since it communicates with the maritime or aeronautical station located outside of the territorial airspace or waters of the coastal state that deploys 5G NR. Figure 5 presents modeling of the antenna patterns of AMS omnidirectional and directional antennas.

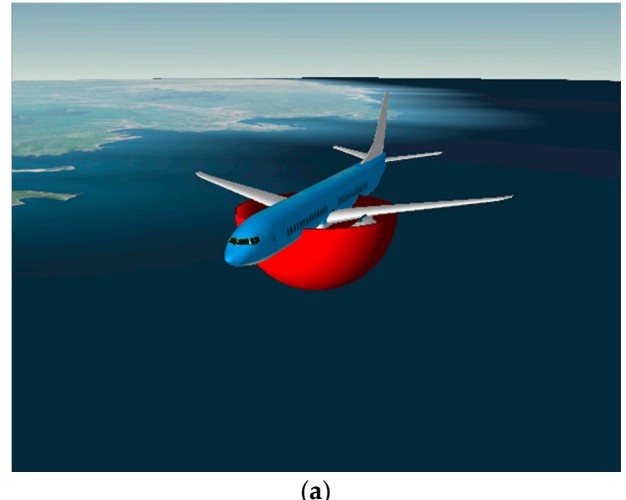 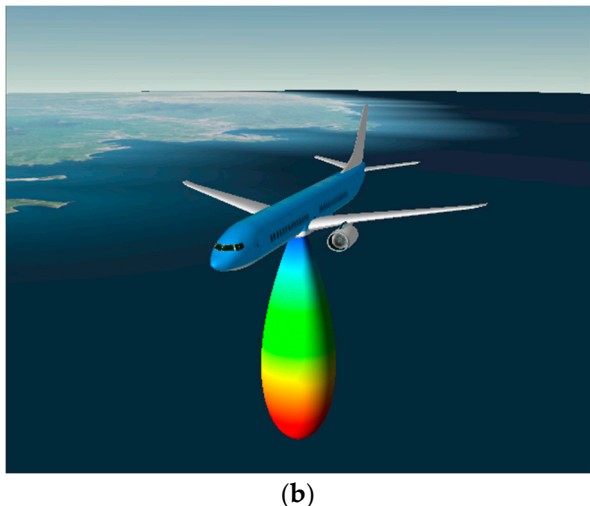

(**a**) (**b**)

**Figure 5.** Antenna pattern of aeronautical receiver: (**a**) omnidirectional antenna of System 1 and (**b**) directional antenna of System 2.

### 3.4. Characteristics Maritime Mobile Service

Two types of MMS receivers were considered in the studies of the one with an omnidirectional antenna (System 1) and the one with a directional antenna (System 2). These parameters are described in Recommendation ITU-R M.2116. Table 6 presents characteristics of the MMS that were used in the simulations.

**Table 6.** Characteristics of the maritime mobile systems used in the simulations.

| Parameter | System 1 Shipborne | System 2 Shipborne |
|---|---|---|
| Tuning range | 4400–4940 | 4800–4990 |
| Power output | 39 | 46 |
| Bandwidth (3 dB) | 5.6/11.3/22.6 | 40/50/60/80/100 (software configurable) |
| Noise figure | 6 | 5 |
| Thermal noise level | −100.5 to −94.5 | −93 . . . −89 |
| Antenna type | Omnidirectional | Directional (steerable, MIMO) |
| Antenna gain | 6/4.2/2.5 | 15 |
| 1st sidelobe | N/A | N/A |
| Polarization | Vertical | Vertical |
| Antenna pattern | N/A | Rec ITU-R F.1336 |

The directional antenna of MMS was pointed to the opposite direction of the coastline since it will communicate with the maritime or aeronautical station located outside of the territorial airspace or waters of the coastal state that deploys 5G NR. Figure 6 presents the modelling of the antenna patterns of MMS omnidirectional and directional antennas.

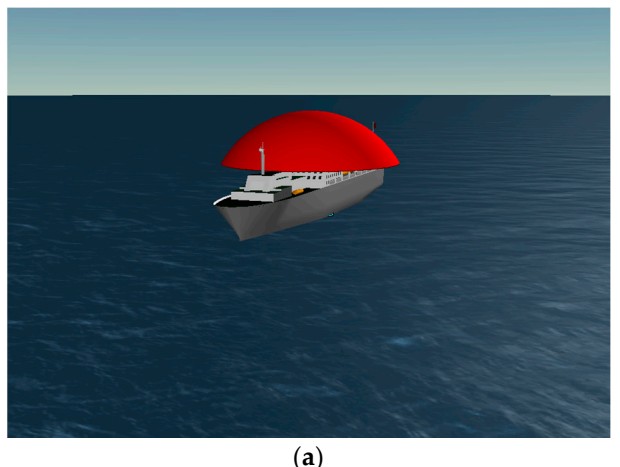
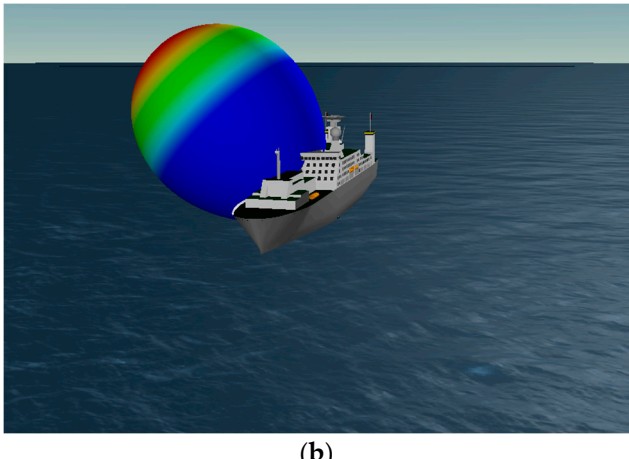

**(a)**                                                                 **(b)**

**Figure 6.** Antenna pattern of maritime receiver: (**a**) omnidirectional antenna of System 1 and (**b**) directional antenna of System 2.

*3.5. Simulation Methodology*

When modeling the simulation, a feeder loss of 2 dB was considered for the AMS and MMS receivers; additionally, 3 dB polarization loss was considered. The altitude of the AMS aircraft was set as 10,000 m, which is a typical average altitude of an aircraft that flies in such areas.

The interference from the ith active IMT-2020 BS or the jth active UE to the FS receiver can be calculated by the following equation [18–20]:

$$I_{IMT} = P_{TX} + G_{IMT} + G_{MS} - L_p - L_{xpr} - L_{feed} \tag{1}$$

where $P_{TX}$ is the transmitted power of the IMT BS or UE, in dBm; $G_{IMT}$ is the transmit antenna gain of the IMT BS or UE towards the victim receiver, in dBi; $G_{MS}$ is the receiver antenna gain of the AMS/MMS towards the interfering station, in dBi; $L_p$ is the propagation loss from the IMT transmitter to the AMS/MMS receiver, in dB; $L_{xpr}$ is the polarization loss, in dB; and $L_{feed}$ is feeder loss of AMS/MMS, in dB. Figure 7 shows 3D modelling of the aircraft or vessel being interfered while moving across the territorial waters border.

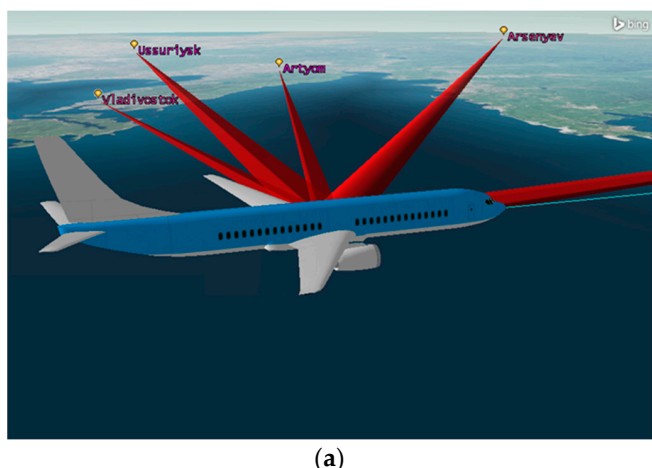
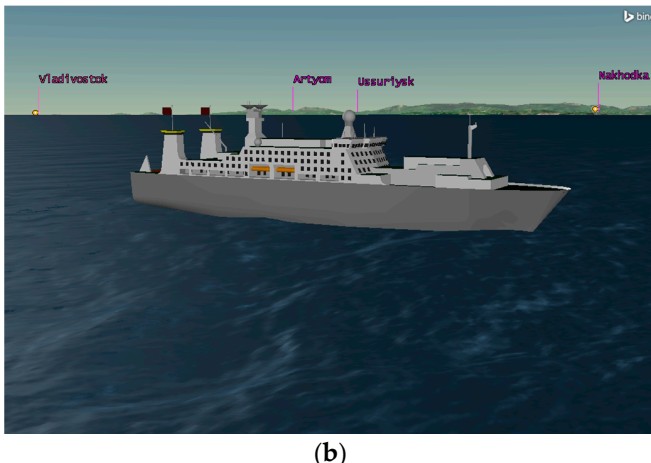

(**a**) (**b**)

**Figure 7.** Scenario for the victim receiver station is located at 12 nautical miles from the coastline: (**a**) aeronautical mobile station at 12 nautical miles from the coastline, and (**b**) maritime mobile station moves along the border of territorial waters.

After the calculation interference from each IMT active station, the total I/N taking into account aggregate interference from IMT-2020 BS and UE can be calculated using the following equation [18–20]:

$$\frac{I}{N}[\text{dB}] = 10log_{10}\left(\text{Pr}_{\text{IMT}} \sum_i 10^{\frac{I_{IMT}(i)}{10}}\right) - (D + 10log(B)) \tag{2}$$

where $I_{IMT}$ the interference from ith active IMT BS or jth active UE to the FS receiver, respectively, in dBm; $D$ is AMS/MMS receiver noise power density, in dBm/Hz; and $B$ is AMS/MMS receiver channel bandwidth.

No downlink power control scheme is applied at the base station and the transmission power per resource block (RB) is constant. For uplink, cell capacity in OFDMA-based systems is constrained by interference levels from other UEs, and in simulations, the UE power control algorithm was used as follows [20,21]:

$$P_{\text{PUSCH}}(i) = \min\left(P_{\text{CMAX}}, 10\log_{10}(M_{\text{PUSCH}}(i)) + P_{\text{O\_PUSCH}}(j) + \alpha(j) \cdot PL\right) \tag{3}$$

where $P_{\text{PUSCH}}$ is transmit power of the terminal in dBm, $P_{\text{CMAX}}$ is maximum transmit power in dBm, $M_{\text{PUSCH}}$ is number of allocated RBs, $P_{0\_PUSCH}$ power per RB is the used target value in dBm, $\alpha$ is a balancing factor for UEs with bad channels and UEs with good channels, and PL is path loss in dB for the UE from its serving BS.

Figure 8 presents a simulation example of the 5G NR networks located in 5 cities for the AMS and MMS stations.

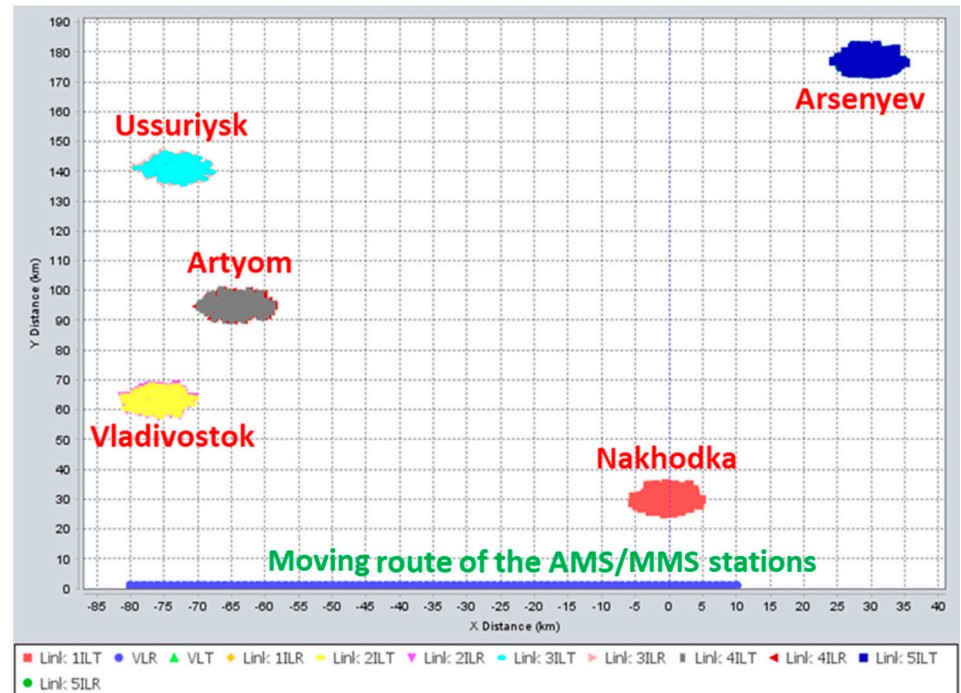

**Figure 8.** Typical 5G NR urban network topology.

In simulations, four propagation models were used:

- Recommendation ITU-R P.528 "A propagation prediction method for aeronautical mobile and radionavigation services using the VHF, UHF and SHF bands" with 20% percentage of time was used to estimate interference with the AMS receivers [22].
- Recommendation ITU-R P.452 "Prediction procedure for the evaluation of interference between stations on the surface of the Earth at frequencies above about 0.1 GHz" with 20% percentage of time, 452 was used to estimate interference level to the MMS receivers [23].
- Recommendation ITU-R P.2108 "Prediction of clutter loss" with 20% of location, the clutter was applied to all 5G interfering BS [24];
- Recommendation ITU-R P.2109 "Building entry loss" with 50% traditional and 50% thermally efficient buildings, this model was applied for the indoor 5G interfering UEs [25].

In addition, for both scenarios, terrain data were used for every interfering link, and Figure 9 shows an example of terrain losses from 1 BS located in Ussuriysk to the maritime station. For AMS, no terrain is required since transmission mode from an airborne station to a region before its radio horizon is the line-of-sight mode, and thus the terrain could barely affect the propagation between the 5G station and AMS and therefore was not considered.

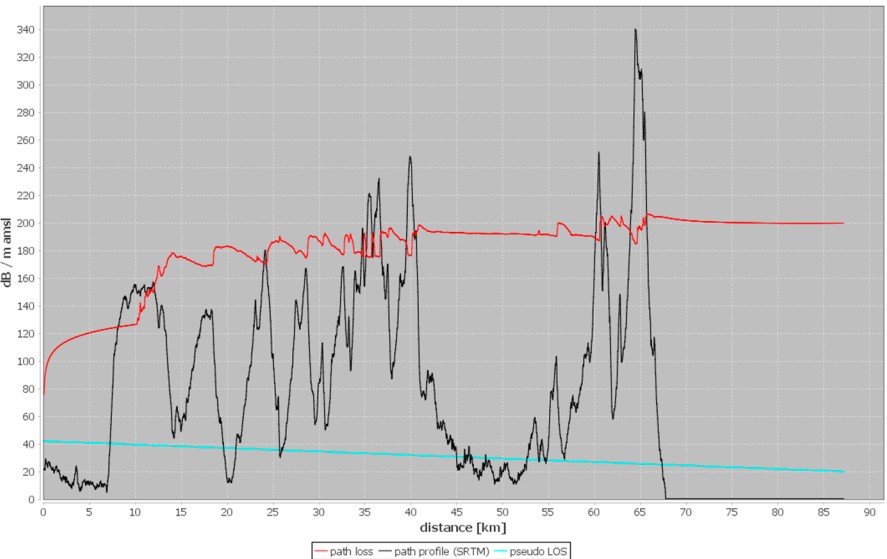

**Figure 9.** Example of path profile between maritime station and one of the IMT-2020 BS in Ussuriysk.

## 4. Results

### 4.1. Results for Aeronautical Mobile Serivce

#### 4.1.1. Results for AMS with Omnidirectional Antenna

Interference analysis for the system with the omnidirectional receiver for different distances showed that I/N protection criteria is met 97% of the time for the worst-case scenario and has more than 14 dB margin for other distances. Figure 10 show distribution functions of I/N when interfering from IMT to the System 1 AMS receiver.

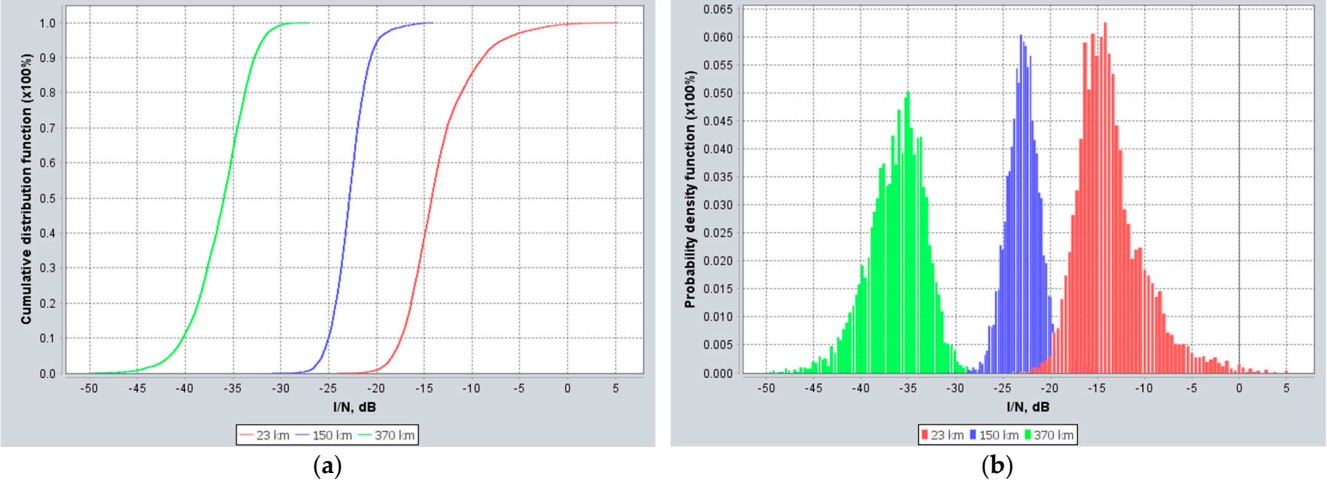

(**a**)                                                                              (**b**)

**Figure 10.** I/N distribution for AMS System 1 with omnidirectional antenna: (**a**) cumulative distribution function of I/N for System 1 AMS receiver and (**b**) probability distribution function of I/N for System 1 AMS receiver.

#### 4.1.2. Results for AMS with Directional Antenna

Interference analysis for the system with the directional receiver for different distances showed that I/N margin is more than 6 dB. Figure 11 shows distribution function of I/N when interfering from IMT to the System 2 AMS receiver.

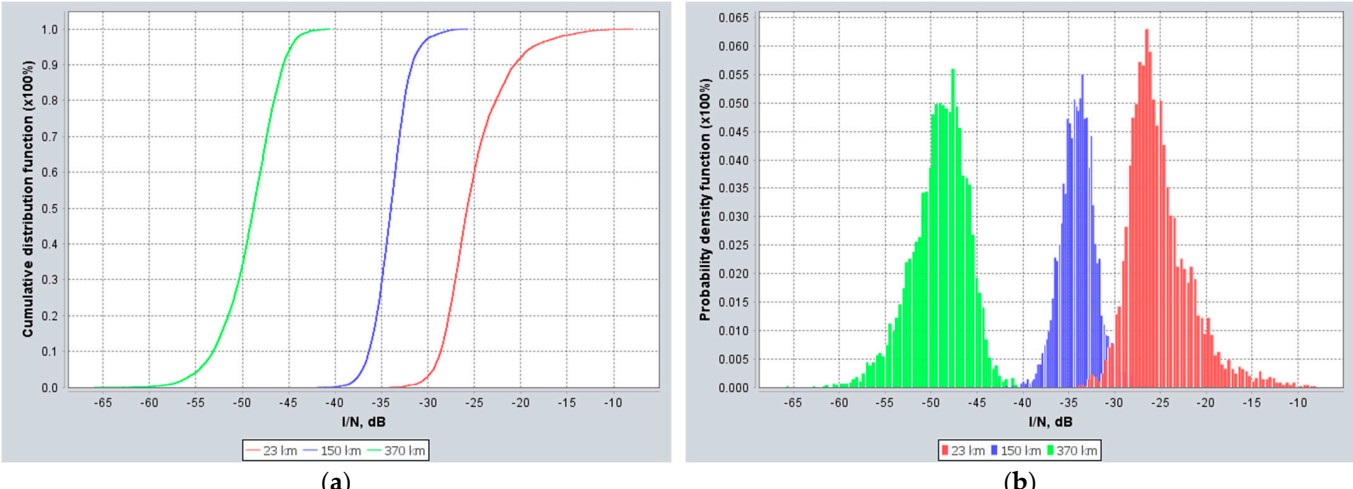

(**a**)                                                                (**b**)

**Figure 11.** I/N distribution for AMS System 2 with directional antenna: (**a**) cumulative distribution function of *I/N* for System 2 AMS receiver and (**b**) probability distribution function of *I/N* for System 2 AMS receiver.

### 4.2. Results for Maritime Mobile Serivce

### 4.2.1. Results for MMS with Omnidirectional Antenna

Interference analysis for the system with the omnidirectional receiver for different distances showed that *I/N* protection criteria is all cases with 7 dB margin. Figure 12 shows distribution functions of *I/N* when interfering from IMT to the System 1 MMS receiver.

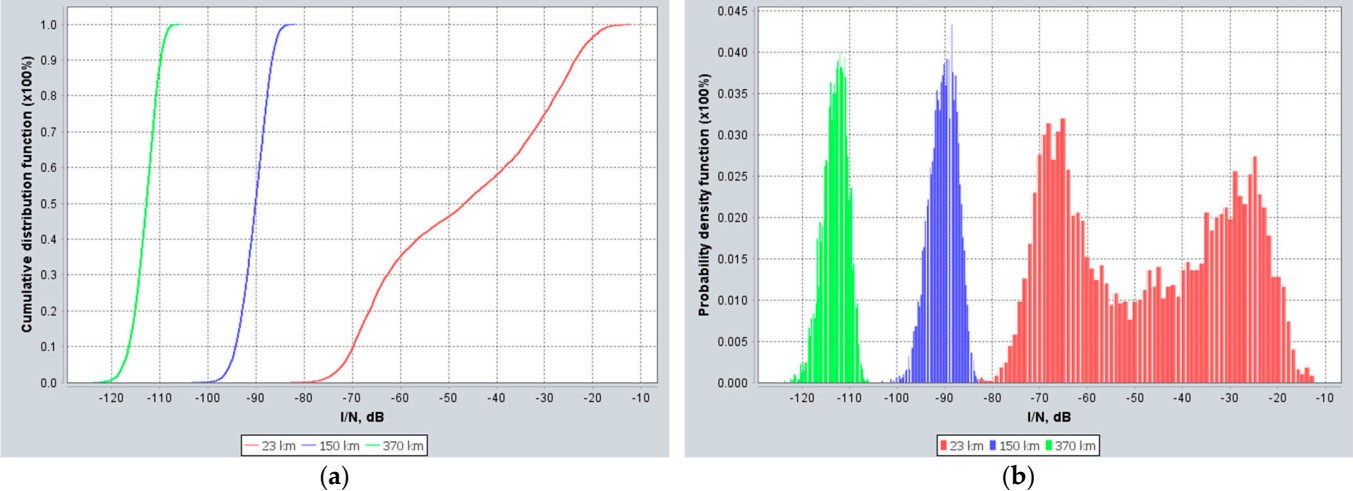

(**a**)                                                                (**b**)

**Figure 12.** I/N distribution for MMS System 1 with omnidirectional antenna: (**a**) cumulative distribution function of *I/N* for System 1 MMS receiver and (**b**) probability distribution function of *I/N* for System 1 MMS receiver.

### 4.2.2. Results for MMS with Directional Antenna

Interference analysis for the system with the omnidirectional receiver for different distances showed that *I/N* protection criteria is all cases with 14 dB margin. Figure 13 shows the distribution functions of *I/N* when interfering from IMT to the System 2 MMS receiver.

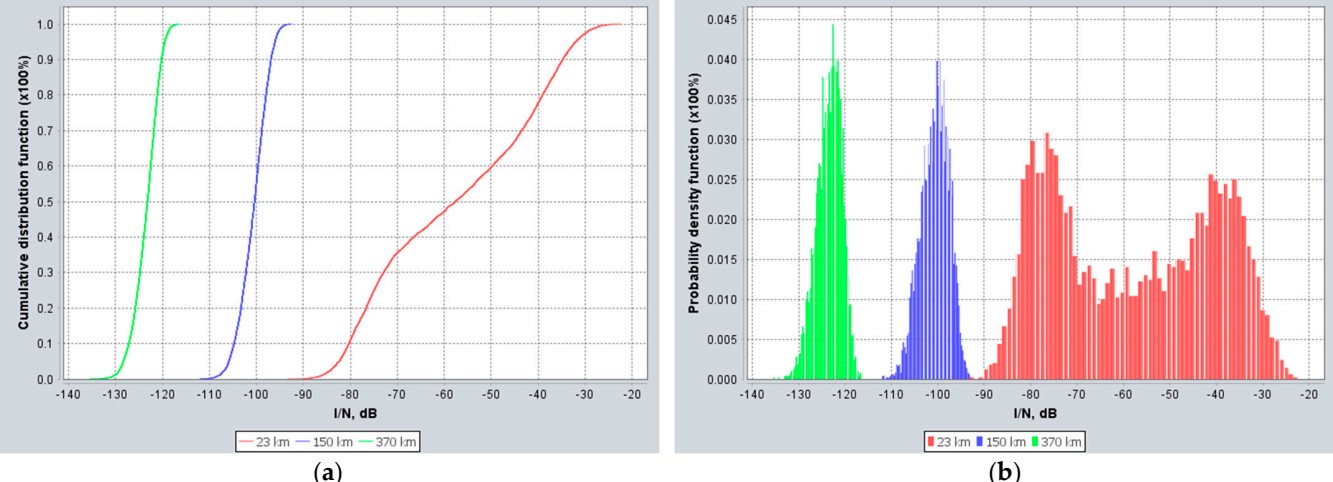

**Figure 13.** *I/N* distribution for MMS System 2 with directional antenna: (**a**) cumulative distribution function of *I/N* for System 2 MMS receiver and (**b**) probability distribution function of *I/N* for System 2 MMS receiver.

## 5. Discussion and Conclusions

The study performed simulations for different types of receivers of aeronautical and maritime mobile services and different distances between the victim receiver and the coastline. It was observed that the results largely depend on these two parameters. At the same time, the results showed that even for the worst-case set of parameters, the interference levels from deployed 5G NR networks to the victim receivers of aeronautical and maritime mobile services did not exceed the protection criterion levels of interference-to-noise ratio, and thus it is not expected that 5G NR deployment at the coastline areas will have an adverse effect on the operation of the aeronautical and maritime mobile services located in international airspace and international waters.

For an AMS receiver with an omnidirectional antenna, the protection criterion is met more than 97% of the time, even in the worst-case scenario when the AMS receiver is located 12 nautical miles from the coastline. For AMS with a directive antenna receiver, the I/N margin for the worst-case scenario is 6 dB. The study assumed that in practice the AMS receiver would be located much farther from the coastline, and the results show that under this assumption the I/N margins are more than 14 dB.

For MMS with an omnidirectional antenna in the worst-case scenario when the MMS receiver is located 12 nautical miles from the coastline, the margin is 7 dB. For an MMS receiver with a directional antenna, the I/N margin for the worst-case scenario is 16 dB. The study further assumed that in practice the MMS receivers would be located much farther from the coastline, and the results show that under this assumption the I/N margins are very significant and equal to more than 90 dB.

Such results indicate that active antenna systems allow mitigating the interference and that 5G NR may be used in the bands where previously it was considered that cellular technologies are not compatible with incumbent services. Additionally, it must be taken into account that any interference from 5G NR will have a fluctuating nature due to the many time-varying parameters of the 5G NR radio link as well as the velocity of the victim vessels or aircraft. Our study showed that, compared to deterministic analysis, stochastic simulations using the Monte Carlo methodology are more precise and closer to the real case scenarios. While the Monte Carlo approach might require more computing power and time, it helps to understand the long-term impact of ubiquitously deployed networks on the incumbent services taking into account time-varying parameters and fluctuations of interference in time.

It should be noted that the interference-to-noise protection criterion is quite stringent and normally used for scenarios where the wanted signal of the victim system varies and

it is difficult to understand which level of the wanted signal should be considered for the studies. At the same time, in practice, these aeronautical and maritime systems have even higher protection margins if measured using a signal-to-interference ratio. Additionally, since most of the aeronautical and maritime mobile systems have broad tuning ranges, in many cases, there is a possibility to avoid interference tuning to the frequency ranges outside of the 4800–4900 MHz frequency band, and such an approach may be used in case the aircraft or vessel approaches the sovereign border of the state that has deployed 5G NR networks. In general, the obtained results of the constructed scenario of the study and simulation model may be applied to other regions since the interference levels will be relatively similar. In most cases, while bigger cities may have more simultaneously active 5G NR equipment in the 4800–49,900 MHz frequency band, bigger cities tend to have higher building heights, and therefore propagation losses due to the clutter shielding will be much better and will compensate the extra amount of interference levels. However, some particular cases may have exceptions and therefore may require further evaluation to understand the conditions of the 5G NR deployment in the coastline areas or maritime service; in particular, the interference levels depend a lot on the terrain region and climate conditions, so in some other regions different results might be obtained. Thus, more experiments and simulations may be needed to understand the long-term effect of the deployed 5G NR base stations at the coastline.

**Author Contributions:** Conceptualization, A.P.; methodology, A.P. and V.S.; software, A.P.; validation, A.P.; formal analysis, V.S.; investigation, A.P. and V.S.; resources, A.P.; data curation, V.S.; writing—original draft preparation, A.P.; writing—review and editing, A.P. and V.S.; visualization, A.P.; supervision, A.P.; project administration, A.P.; funding acquisition, A.P. All authors have read and agreed to the published version of the manuscript.

**Funding:** This research received no external funding.

**Institutional Review Board Statement:** Not applicable.

**Informed Consent Statement:** Not applicable.

**Data Availability Statement:** Not applicable.

**Acknowledgments:** The authors would like to express thanks to the Radio Research and Development Institute (NIIR) for its extensive support and consultations.

**Conflicts of Interest:** The authors declare no conflict of interest.

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
