# Peer review of "Evaluation of Interference Analysis from 5G NR Networks to Aeronautical and Maritime Mobile Systems in the Frequency Band 4800–4990 MHz"

_2571-8800, doi:10.3390/j6010002_

Round 1

Reviewer 1 Report

 Evaluation of interference analysis from 5G NR networks to 2 aeronautical and maritime mobile systems in the frequency 3 band 4800-4990 MHz

After reviewing he above research paper, the following modifications are required

1.      List the novelties of the current design before the material and methods before section 2

2.      Improve the quality of the graphs throughout the paper

3.      Include the 5G NR frequency bands in the paper

4.      Assign numbers for equations in the research paper

5.      What is the basic expression for the design of current design for the applications of 5G NR bands

6.      Add some more recent papers in the reference section related to the current research paper better to understand the current topic.

Author Response

Thank you for your review!

  1. Added the novelties which in a newly developed section 2.
  2. Unfortunately, it’s not possible to improve it, the research was done using SEAMCAT tool which is designed for spectrum engineering analysis using Monte-Carlo method. And this style of graphs is part of the software interface, which is not possible to change. But actually in the word version the quality of the graphs is fine, I think converting to pdf properly would help to improve the quality.
  3. Added 5G NR bands in the introduction section.
  4. Added numbers to the equations
  5. Added the applications of 5G NR in the introduction section.
  6. The topic is quite new, and I would say that we are probably the first who publish the work in that topic. I have added some links to the studies that are done within International Telecommunication Union this year https://www.itu.int/dms_ties/itu-r/md/19/wp5d/c/R19-WP5D-C-1555!H4-N4.08!MSW-E.docx our study was also considered by the working part 5D of ITU-R and added to these supporting materials. In addition, here’s a GSMA report that describes briefly the history of the issue https://www.gsma.com/spectrum/wp-content/uploads/2021/04/WRC-23-IMT-Agenda-Items.pdf

Reviewer 2 Report

This  study based on the example of the Pacific region where a simulation of aggregate interference from 5G NR base stations and user equipment deployed in the cities near the coastline using Monte-Carlo analysis was conducted. The results of the study show that no harmful interference to the aeronautical and maritime services operating in the international airspace. The results need more experiments.

Author Response

In the introductory part, we have specified that it’s not possible to do the experiments. The reason for that is that because we are dealing with aggregate interference from the base stations deployed within large areas. Both maritime and aeronautical stations will receive interference from hundreds of base stations located near the coastlines. It’s not possible to conduct an experiment where you place hundreds of base stations in the coastline and then check whether the aircraft or a vessel that moves 20-100 km away of the coastline will receive significant amount of interference. And it’s also not possible to do such experiment in the lab as well. That’s one of the challenges in this task, that before the actual deployment it’s possible to do only theoretical studies.

What I can do is that add to the conclusion section that the obtained results may require more detailed investigation and more theoretical studies that more detailed evaluate the density of base stations is required. That would be to conclude that more experiments are required. 

Reviewer 3 Report

The novelty of the paper should be explained more. Why are you focusing on the frequency band 4.8kHz-4.99kHz?

One of the biggest deficiencies of the paper is that the mathematical background (Evaluation of interference analysis) is missing. The paper must be improved in this case.

The introduction does not provide a comprehensive overview of the topic, and there are insufficient references. For this kind of paper, 20 references used in the paper are not sufficient. here are a few suggestions for reference. Authors may add them in the introduction as well.Illahi, U., et al. "Bandwidth Enhancement of Rectangular Dielectric Resonator Antenna with and Without a Parasitic Patch." Journal of Engineering Technology 5 (2017): 5-8, Zambak, M.F.; et al." Higher-Order-Mode Triple Band Circularly Polarized Rectangular Dielectric Resonator Antenna". Appl. Sci. 2021, 11, 3493. https://doi.org/10.3390/app11083493,J. Iqbal, Usman Illahi, M.N.M. Yasin, Mahmoud A. Albreem, M.F. Akbar, Bandwidth enhancement by using parasitic patch on dielectric resonator antenna for sub-6 GHz 5G NR bands application, Alexandria Engineering Journal, 2021. 

The Conclusion part of the paper is too short and the deeper scientific discussion about the achieved results is missing.

Author Response

The reason why the study focuses on this band is that unlike most of the other bands, this band has potential compatibility problems with incumbent services. And this issue is considered within the ITU-R.

I have updated suggested references, and added one more, so overall there are now 25 references.

The conclusion part was updated with additional discussions.